# Determining the Optical Geometry of a Gold Semi-Shell under the Kretschmann Configuration

**Tomoki Watanabe** [1,*] **, Toshihiro Okamoto** [1,2]**, Kenzo Yamaguchi** [1,2] **and Masanobu Haraguchi** [1,2]

1   Graduate School of Advanced Technology and Science, Tokushima University, 2-1, Minami-Josanjima, Tokushima 770-8506, Tokushima, Japan; toshi-okamoto@tokushima-u.ac.jp (T.O.); yamaguchi.kenzo@tokushima-u.ac.jp (K.Y.); haraguchi.masanobu@tokushima-u.ac.jp (M.H.)
2   Institute of Post-LED Photonics (pLED), Tokushima University, 2-1, Minami-Josanjima, Tokushima 770-8506, Tokushima, Japan
*   Correspondence: c502148002@tokushima-u.ac.jp

**Abstract:** Dielectric nanoparticles coated with metals (half-shell or semi-shell structures) have attracted attention as potential composite plasmonic nanomaterials with large optical anisotropy and absorption cross-sections. Structures approximately 100 nm in size can excite plasmons in the visible and near-infrared ranges, highlighting their distinct optical properties. This study employed metal semi-shell structures (metal: gold, dielectric: silica) in the Kretschmann configuration to experimentally and numerically demonstrate the optical determination of single-structure orientations through a finite-difference time-domain method. Gold semi-shell structures were fabricated through deposition and etching. These structures were removed from their substrate in ultrapure water and randomly dropped onto a thin gold substrate. In the single structure, we experimentally observed changes in the scattering light spectrum based on the optical geometry of the gold semi-shell at wavelengths ranging from 530 to 700 nm. The obtained results closely resembled those of a simulation and confirmed the presence of eigenmodes in the orientation through electric field analysis. These observations allow for the cost-effective and rapid determination of the orientations of numerous structures that are approximately 100 nm in size, solely through optical methods. This technique is a valuable development for measurement applications in nanostructure orientation control and functionality enhancement.

**Keywords:** Kretschmann configuration; plasmonics; optical geometry; semi-shell scattered property; scattered-light spectroscopy

## 1. Introduction

The excitation of surface plasmons on metal surfaces induces distinct optical properties in plasmonic nanostructures [1,2]. Recently, plasmonic nanostructures have been used in diverse applications, including as metamaterials [3–8], and for nonlinear optical effects [9,10], optothermal and photoelectric conversion [11,12], and biosensing [13–16]. Notably, surface plasmon resonance strongly depends on the shape, size, material, and surrounding media of nanoparticles [1,2]. Consequently, plasmonic nanostructures have been designed with various shapes, such as nanorods [3,4], fishnet structures [5,6], and nanohole arrays [7,8]. In particular, half-shell and semi-shell structures involving dielectric nanoparticles coated with metals are attracting attention as composite materials with optical anisotropy and large absorption cross-sectional areas [17–19]. Figure 1 illustrates the concepts of half-shell and semi-shell structures. A half-shell structure is a structure in which half of the nanoparticle is coated with metal, forming a cap-shaped configuration. Conversely, a semi-shell structure, as addressed in this study, is defined as an asymmetrical ring structure coating a nanoparticle. These structures can couple light with typically forbidden plasmon modes [18]. Each individual nanoparticle can be utilized as a material with unique optical properties. Unlike a half-shell structure, semi-shell structures,

being asymmetric rings, can generate a magnetic dipole response to incident light [18]. Using nanoparticles as sources, the resonance wavelength can be adjusted by changing the particle size. Reducing the size to approximately 100 nm can enable the excitation of plasmon modes in the optical wavelength range (visible and near-infrared regions). In this specific optical wavelength range, these responsive structures can be applied to fluorescent biosensing and metamaterial optical devices. Therefore, these materials are advantageous for realizing functions within the optical wavelength range.

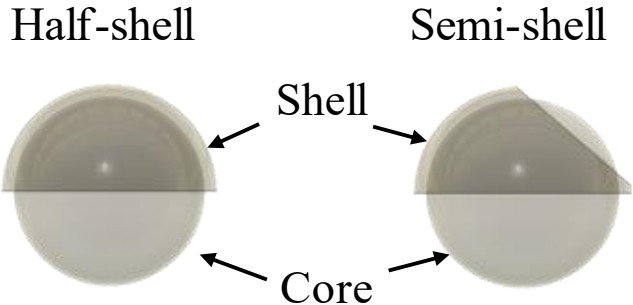

**Figure 1.** Shell structure concepts.

The nano-sphere lithography (NSL) method [20] is used to mass-produce half-shell and semi-shell structures, which can be removed from the solution as water dispersions [21]. We anticipate the ability to add functionality as a two-dimensional metamaterial to various substrates, including those with curved surfaces, by employing semi-shell dispersion. However, since the arrangement and orientation of semi-shell structures tend to be random, it is difficult to fully harness their potential as metamaterials. Research is being conducted on techniques to control the arrangement and orientation of semi-shell structures, and observing the orientation of the semi-shell structures is crucial for realizing these techniques. To make semi-shell structures responsive in the visible and near-infrared range, they need to be around 100 nm in size. High-resolution SEM is a powerful method for observing the orientation of objects of this size. However, it is not great for dielectric materials. That is because it uses electron beams for observation, and dielectric materials tend to gather and hold onto electric charge, making them prone to charge-up effects. The semi-shell structures used in our study are quite small, about 100 nm in diameter, and this makes observation even more difficult due to the likelihood of charge-up effects. Therefore, there is a need for a simple and quick method to measure orientation. Optical methods offer a promising approach in this regard.

Optical measurement techniques at the nanoscale include the use of surface plasmon resonance (SPR) [22–25]. To excite surface plasmons, it is necessary to employ techniques such as the attenuated total reflection (ATR) method or employ strategies like coupling wavevectors using diffraction gratings. An exemplary arrangement is the Kretschmann configuration, which consists of three layers: a high-refractive-index dielectric, a thin metal film, and a low-refractive-index dielectric [26]. In this configuration, when light is incident from the high-refractive-index medium at an angle beyond the critical angle, it generates evanescent light on the surface of the metal film facing the low-refractive-index dielectric. The resulting evanescent light excites surface plasmons. In recent years, numerous measurement techniques using SPR have been developed, particularly in the field of bioscience. These include studies that combine SPR with photonic crystal structures to achieve enhanced sensitivity through energy confinement at specific wavelengths, resulting in an increased fluorescence signal intensity [27]. There have also been investigations into methods that use SPR to increase the transmittance in relation to the field associated with evanescent waves generated by subwavelength-sized metal hole structures [28,29]. SPR is most important when it comes to acquiring signals dependent on the orientation of a single semi-shell structure of approximately 100 nm for a specific direction. However, when using photonic crystal structures, the enhancement effect is limited to specific wavelengths. In

the case of utilizing periodic metal structures, one must consider the location-dependent variations resulting from structural errors in both the metal and the periodic structure. Hence, we have chosen to employ SPR in the Kretschmann configuration, which features a flat metal surface and exhibits enhancement effects across a wide range of wavelengths. Surface plasmons occur when light resonates at specific wavelengths or energies on a metal surface, confining the electromagnetic field spread to the metal surface. As a result, structures located near the metal thin film receive a significant amount of energy. Therefore, variations in the orientation of the structure lead to differences in the distance between the metal thin film and the metal structure of the semi-shell. Depending on these distances, changes in the characteristic signals are expected to be particularly pronounced owing to shifts in the coupling modes.

In this study, we successfully fabricated numerous semi-shell structures with asymmetric openings using NSL [20] and extracted them as water dispersions. The semi-shell structures were arranged in a Kretschmann configuration on a separate substrate using a dispersion liquid. This substrate was prepared using a 50 nm thick gold deposition on glass. Notably, surface plasmons enable higher-sensitivity signal detection than conventional microscopy. The scattered-light spectra based on the optical geometric orientations of single structures were recorded using this measurement system. Consequently, we demonstrated that the orientations of single structures could be determined solely through optical methods by comparing finite-difference time-domain (FDTD) numerical calculations with experimentally obtained scattered-light spectrum data.

## 2. Materials and Methods

### 2.1. Gold Semi-Shell Dispersion Liquid Fabrication

The core and shell of the semi-shell structure were composed of silica particles (Fuji Chemical Inc., Osaka, Japan; Silbol-S150) and gold, respectively. Positively charged 150 nm silica particles were densely dispersed on a glass substrate using the electrostatic adsorption method [30,31] (Figure 2a). The poly(cationic) solution and poly(anionic) solution used in the electrostatic adsorption method were adjusted to contain 1 wt.% and 5 wt.% of {Poly(diallyl methylammonium chloride): PDDA} and {Poly(styrene sulfonate): PSS}, respectively, using MilliQ water as the solvent. Immersion for 10 min, followed by rinsing for 5 min, was performed to prepare the entire substrate for the easy adsorption of silica particles. Optical microscopy observations revealed the presence of approximately 11.82 million individual silica nanoparticles on an 18 mm × 18 mm glass substrate. Gold was thermally evaporated onto the glass substrate at a 45° angle from the vertical direction (Figure 2b). Subsequently, the gold deposited on top of the particles and on the substrate was subjected to argon-ion etching along the vertical direction of the substrate. Typically, etching gold onto a substrate induces a sputtering effect. The sputtering effect from the substrate causes gold to adhere to areas of a nanoparticle to which it does not naturally attach, owing to the influence of the deposition angle. A combination of these effects leads to the formation of gold semi-shell structures with asymmetric configurations around the nanoparticles (Figure 2c). In our study, a gold semi-shell dispersion was obtained by subjecting the substrate to ultrasonic vibration in ultrapure water (Figure 2d). Through ultrasonic vibration, the semi-shell structures were removed from the substrate. In the liquid, apart from the semi-shell structures, other structures were present. These included silica nanoparticles that adhered to the backside of the substrate and silica spheres that remained in areas in which gold was not deposited due to the deposition process. As a result, the liquid contained a mixture of semi-shell structures with metal components and silica particles without metal components. To separate these two types of nanospheres, a centrifuge was utilized to obtain a dispersion of only the semi-shell structures (Figure 2e). This dispersion was then dropped onto a glass substrate with a 50 nm gold thin-film deposition. The sample substrate was left to stand at room temperature for 12 h and then dried using nitrogen blowing, resulting in the preparation of the sample substrate.

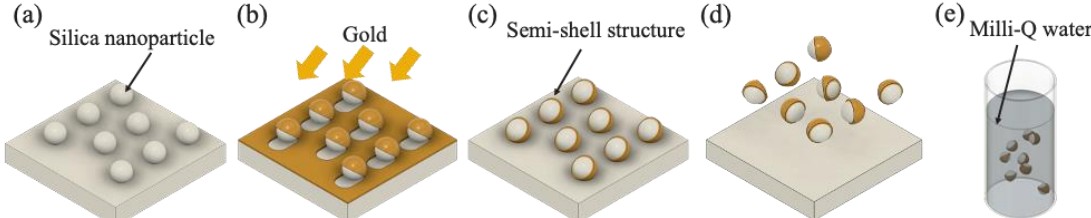

**Figure 2.** Schematic of the gold semi-shell structure dispersion liquid fabrication process. (**a**) Dispersion of silica particles onto an $SiO_2$ substrate via electrostatic adsorption. (**b**) Gold vapor deposition (50 nm) at an angle of 45°. (**c**) Gold semi-shell formation via argon etching and sputtering. (**d**) Removal of structures from the substrate through ultrasonic vibration. (**e**) Separation and extraction of the gold semi-shell dispersion through centrifugation.

### 2.2. Experimental Setup

To evaluate the orientation of the gold semi-shell structures optically, we performed scattered-light spectrum measurements using propagating surface plasmons. Figure 3a shows a schematic of the measurement system. The light source was a supercontinuum (NKT Photonics Co. (Boston, MA, USA)) that emitted strong and coherent light over a wide wavelength range. We polarized the light with p-polarization using a polarizer. Additionally, to further increase the light intensity, we utilized a condenser lens with a focal length of 250 mm for light focusing. The glass surface and prism were optically integrated into a single unit. The p-polarized light was made incident from the prism side at an angle $\theta_i$ onto the gold thin-film substrate placed on the glass surface. The evanescent light generated on the gold thin-film substrate was incident on the gold semi-shell structure. The scattered-light spectrum from the gold semi-shell structure was evaluated from the gold thin-film side. The scattered light from the structure was observed without passing it through a polarizer. Figure 3b shows the spectrum of the electric field component perpendicular to the surface of the evanescent light present on the gold thin-film surface (in the z direction). The red and blue lines correspond to incidence angles of 45° and 70°, respectively. The wavelength spectrum of evanescent light exhibits a strong bias owing to the excitation of propagating surface plasmons. Understanding the orientation of the structure was crucial in this experiment. It is noticeable that the results at an incidence angle of 70° exhibit less wavelength dependence between 500 and 900 nm than the results at an incidence angle of 45°. At an incidence angle of 45°, the wavelength dependence of the penetration depth is significant, resulting in a substantial difference in sensitivity depending on the wavelength. This makes it unsuitable for obtaining the spectral characteristics of the nanoparticles. On the other hand, using a 70° incidence angle reduces the sensitivity differences between wavelengths, mitigating this issue. Therefore, for wide-band spectrum observations, we conducted the experiment with a 70° incidence angle. The proposed method relies on the evanescent wave penetration depth, which determines the size limit for orientation measurement. The penetration depth $Z$ is defined by the following equation:

$$Z = \frac{\lambda}{2\pi\sqrt{(n_1 sin\theta)^2 - (n_2)^2}} \tag{1}$$

Here, $\lambda$ represents the incident wavelength, $\theta$ is the incident angle, $n_1$ is the refractive index of the prism, and $n_2$ is the refractive index of the metal thin film. In our experiment, $Z$ was theoretically expected to have a width of around 1 μm.

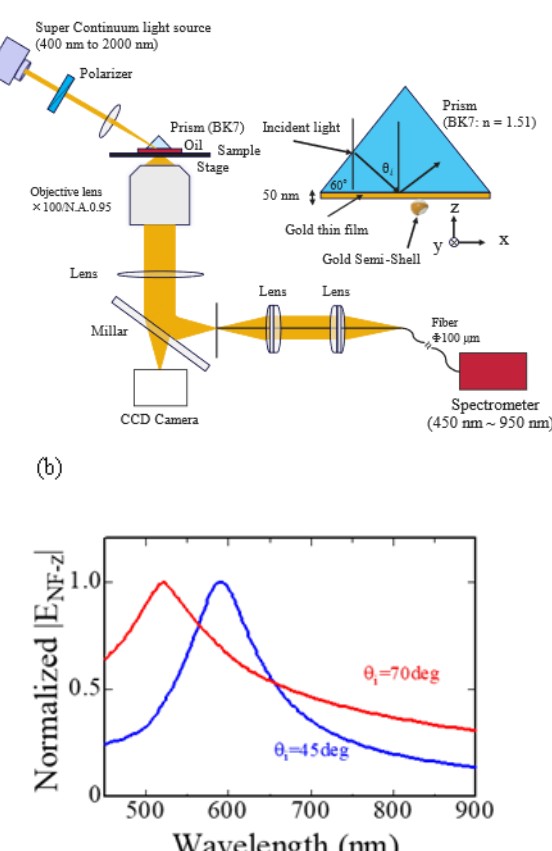

**Figure 3.** Schematic of the measurement system. (**a**) Optical system. (**b**) Surface plasmon excitation condition dependent on the angle of incidence.

The spectrometer (Acton Research Corp. (Acton, MA, USA)) could measure in the range of 450–950 nm, with a spectral resolution of 4 nm. Since we used the Kretschmann configuration for measurements, we could significantly reduce the background light detected in the observation system from sources other than the measurement target. Furthermore, we used the following equation to calculate the scattered light intensity spectrum:

$$\text{Scattering light intensity} = \text{Signal} - \text{Background/Reference} \tag{2}$$

Here, Signal is defined as the scattered-light spectrum from the structure, Background is the scattered-light spectrum from the structure-free gold thin film, and Reference is the spectrum of the light source. Therefore, sufficient background suppression measures were considered.

### 2.3. Numerical Calculation

The electromagnetic field distribution and scattered-light spectra of the gold semi-shell structures were determined using the three-dimensional (3D) FDTD method. Figure 4 shows a model of the simulated structure. The mesh size was set to 2 nm for each of the $x$-axis, $y$-axis, and $z$-axis. Perfectly Matched Layer (PML) boundaries were configured for each of them. The refractive index of the silica particles was 1.5, and their diameter was 150 nm. The refractive index of the prism was assumed to be 1.51. The semi-shell parameters in the simulation model were obtained from an SEM observation image of a semi-shell structure fabricated with a silica particle 500 nm in diameter, as shown in Figure 4a. Based on observations and considering the scale when fabricated with 150 nm particles, it was estimated that the long- ($h_1$) and short-side ($h_2$) widths of the split-ring structure were approximately 70 and 20 nm, respectively. The distance $g$ between the metal structure (metal ring) and the metal thin film was assumed to be equivalent to the

height of the remaining metal split-ring structure on the substrate, using silica particles as a mask. This deposition height depends on the amount of deposition; therefore, even for the semi-shell structures with 150 nm particles, *g* was set to 40 nm. The gap between the ring and gold thin film was set to 40 nm. The dielectric constant of gold was fitted using experimentally obtained values to match the dispersion relation in the wavelength range of 0.188–1.937 μm [32]. The scattered-light intensity was calculated at point *m*, located 1.0 μm away from the center of the silica particle along the direction perpendicular to the substrate. The electric field component was observed in the direction parallel to the incident light (z-direction).

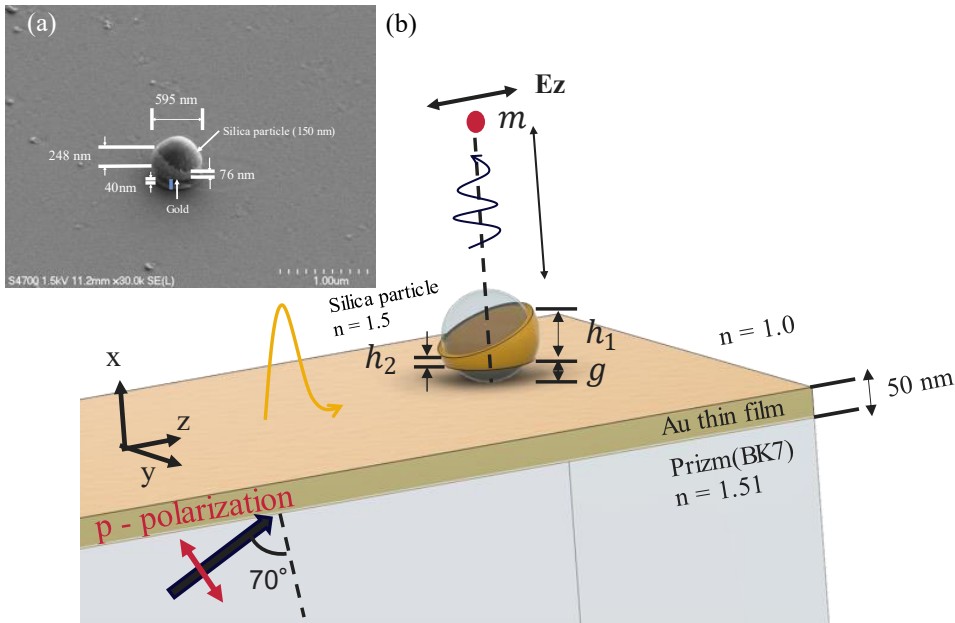

**Figure 4.** Calculation model for the FDTD simulation. (**a**) Semi-shell structure prepared for observation. (**b**) The width of the ring part of the gold semi-shell structure is longer on one side, $h_1$ (70 nm), and shorter on the other side, $h_2$ (20 nm). *g* represents the distance between the gold semi-shell structure and the gold thin film. Point *m* indicates the numerical observation position for the polarization electric field intensity $E_z$.

## 3. Results and Discussion

Figure 5a shows a dark-field microscopy image of the target sample captured using the proposed optical setup. Figure 5b shows the SEM image within the white box of Figure 5a. Overall, the SEM observations confirmed that the target sample consisted of a single particle. The scattered-light spectrum changed depending on the structural geometries of the gold thin-film and gold semi-shell structures when exposed to evanescent light with a perpendicular electric field.

The obtained scattered-light spectrum is shown in Figure 6a. The scattered-light spectrum was recorded and normalized in the wavelength range of 500–950 nm. Sample 001 had a scattered-light intensity peak (P1) at 570 nm that gradually decreased toward 800 nm. Sample 002 exhibited a slight peak (P2) at 600 nm and an exponential decrease in its intensity at 800 nm. Sample 003 clearly exhibited peaks (P3 and P4) at 570 and 630 nm, with a slight peak (P5) observed at 700 nm. Although peaks corresponding to silica particles were observed, the spectrum of silica particles exhibited a smaller signal compared with that in the presence of gold semi-shell structures. The peak intensity of silica particles was approximately 30% that of the gold semi-shell structure. This significantly smaller intensity indicates that the structure corresponded to individual silica particles.

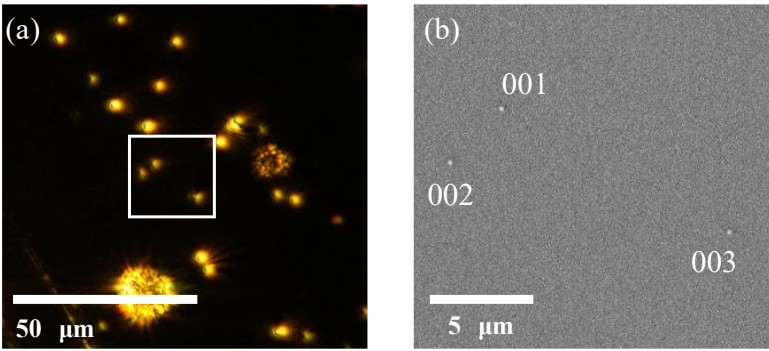

**Figure 5.** Gold semi-shell structures placed on a gold thin-film substrate. (**a**) Dark-field microscopy image. (**b**) The SEM image corresponds to the region within the white rectangle of (**a**).

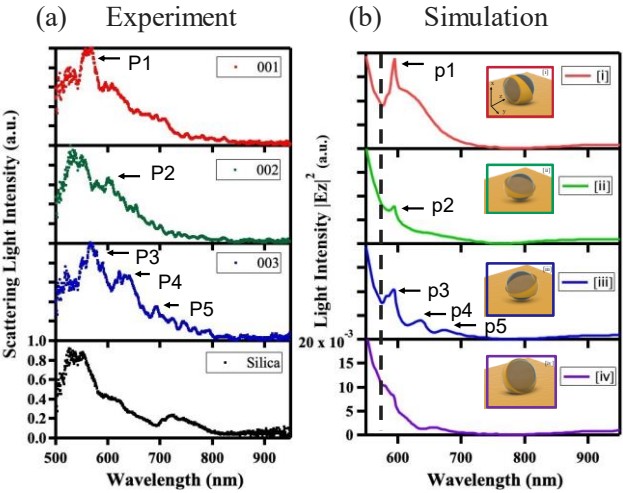

**Figure 6.** Scattered-light spectra of gold semi-shell structures recorded in the (**a**) Kretschmann configuration and (**b**) FDTD simulation results. (**a**) Spectra obtained from sample numbers 001, 002, and 003 are represented by the red, green, and blue lines, respectively, whereas the spectrum of the silica particle is represented by the black line. (**b**) FDTD simulation results for configurations [i–iv] of the gold semi-shell structures.

The optical orientation of the structure, however, appears unclear in the SEM images. Accordingly, the recorded spectra were compared with the simulation results obtained using the FDTD method to determine the orientation of the structure. Figure 6b shows the simulation results corresponding to different arrangements of the gold semi-shell structures [i–iv]. The obtained arrangements comprised four configurations. The excited surface plasmon propagated in the z-direction and incident to the structure. Configuration [i] had a gold semi-shell structure perpendicular to the gold thin-film substrate and was oriented vertically along the z-direction. The configuration was set such that the side with the smaller ring width faced the +x-direction. In configuration [ii], the gold semi-shell structure was oriented horizontally relative to the gold thin-film substrate, with the side having a smaller ring width facing the +y-direction. In configuration [iii], similar to configuration [ii], the gold semi-shell structure was oriented horizontally relative to the substrate, with the side having a smaller ring width facing the −z-direction. In configuration [iv], the gold semi-shell structure was perpendicular to the gold thin-film substrate and oriented horizontally along the +z-direction.

Figure 6b[i] shows a spectrum with a sharp peak (p1) at approximately 600 nm, followed by a broad decrease in intensity up to 750 nm. Figure 6b[ii] shows a peak (p2) around 600 nm, which is similar to that in Figure 6b[i] but subtle. The scattered-light intensity decreases exponentially above 600 nm. Figure 6b[iii] shows three peaks (p3–p5)

in the range of 550–700 nm. Unlike the other three configurations, the configuration in Figure 6b[i] does not exhibit a peak around 600 nm.

The modes associated with these peaks are discussed below using the electric field distribution diagrams shown in Figure 7. Figure 7a,f shows the electric field distribution diagrams of the $E_x$ and $E_z$ components for p1 at a wavelength of 595 nm. The terms A1 and A2 in Figure 7 correspond to the heights $h_1$ and $h_2$, respectively, of the gold rings, as indicated in Figure 3b. At A1 and A2, as depicted in Figure 7a, the strength of the electric field increases. In Figure 7f, the A1 portion indicates a strong electric field intensity. The proximity to the gold thin film makes the ring part more susceptible to the enhancement effect of surface plasmons. Thus, the spectrum peak at a wavelength of 595 nm in configuration [i] is believed to be caused by the presence of the ring portion, with height $h_1$ located on the gold thin-film side. This characteristic is also evident in the signal depicted in Figure 6a, P1 of No. 001. By contrast, in configuration [ii], the gold ring structure is oriented horizontally relative to the gold thin film and exhibits a symmetric arrangement along the z-direction. From Figure 7b,g, the presence of symmetry can be confirmed. Owing to this symmetry of the structure, the coupling with plasmon modes is weak. Additionally, the distance from the gold thin film is increased. Therefore, p2 exhibits a lower peak electric field intensity than p1, as also observed in the signal depicted in Figure 5b, P2 of No. 002.

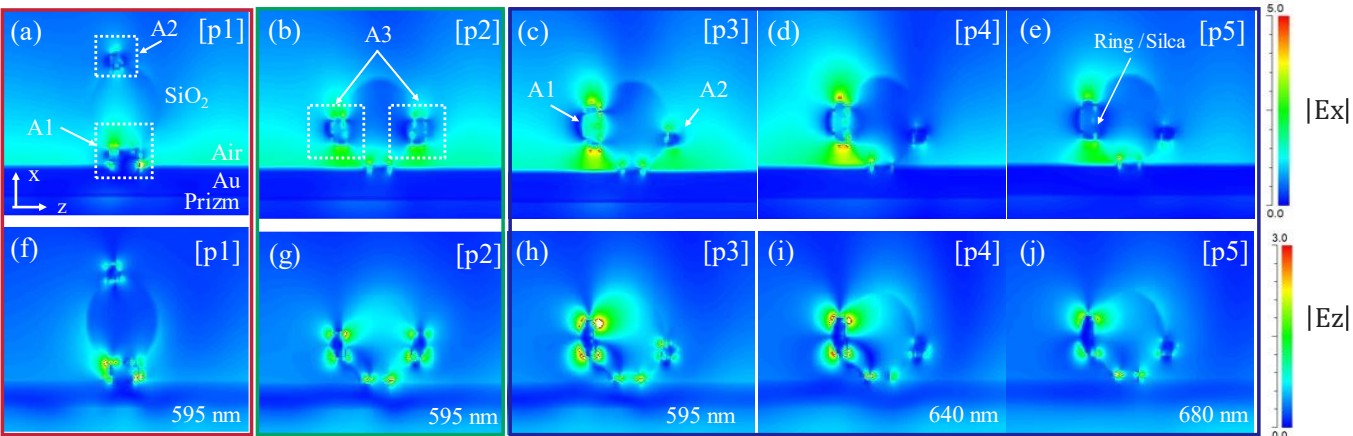

**Figure 7.** Electric field distribution diagram of the $E_x$ and $E_z$ components in the x–z plane at each wavelength for the spectral peaks. (**a**,**f**) [i] configuration (λ = 595 nm). (**b**,**g**) [ii] configuration. (**c**–**e**) and (**h**–**j**) [iii] configuration (λ = 595 nm, 640 nm, and 680 nm).

Figure 7c–e,h–j shows the electric field intensity distribution at each peak wavelength; note that Figure 7h–j does not exhibit significant differences; however, Figure 7c–e shows differences in distribution. In (c), the electric field is strong in parts A1 and A2, owing to the coupling of different modes arising from these two locations. In (d), no significant electric field intensity is observed for the part with A2. Thus, the peak at p4 is likely dominated by the mode originating from the A1 part. In (e), the electric field intensity at the interface between the ring portion and silica particle is lower compared with that in (c) and (d). This indicates that the modes dominating at the interface between the ring portion and air are more pronounced. These characteristics are also confirmed in the signal depicted in Figure 5a, P3–5 of No. 003. The variations in the electric field distribution change the configurations; these differences impact the spectrum of the electric field intensity.

A comparison of the simulation and experimental results based on the spectral profiles reveals that the sample numbers 001, 002, and 003 correspond to configurations [i], [ii], and [iii], respectively. Both the experimental and simulation data reveal that the spectral intensity decreases with increasing wavelength on the longer wavelength side. This behavior is attributed to the wavelength dependence of the intensity of evanescent light, as shown in Figure 2b, which is influenced by the incident light angle. Below 580 nm, the

simulated light intensity spectrum increases with decreasing wavelength, whereas the experimental intensity data decreased above 530 nm. This difference is attributed to an unaccounted decrease in the transmittance of the gold thin film below 580 nm [33,34] that was observed in the experiment, which is not reflected in the simulation. Additionally, the peak wavelengths of the experimental and simulated results differ. This difference is attributed to the variation in the gold coverage ratio and shape of the actual fabricated structure compared to those in the simulation model.

## 4. Conclusions

Here, we established a measurement method using surface plasmons based on the Kretschmann configuration to determine the geometrical orientation of a gold semi-shell structure on a gold thin film using only optical measurements. This method enables the determination of the orientation of structures, which is challenging to observe using conventional techniques owing to material limitations. Moreover, because the proposed method is an optical measurement, it can be performed without damaging the sample, and rapid orientation determination is possible solely based on spectral data. This method, when using different metals, results in a change in the resonance wavelength owing to the variation in metals. However, similar to when gold is used, it is expected to provide characteristics based on the orientation of the structure. Therefore, this measurement technique remains highly versatile and can be applied to other metals as well. We conducted measurements only in the visible range; by utilizing the proposed method to obtain high-sensitivity information in the near-infrared range, the orientation of the structure can be determined with greater precision. This improved sensitivity in the near-infrared region can provide valuable insights into the characteristics of nanostructures, such as gold semi-shell structures. Thus, our proposed method may have various applications, including in imaging, sensing, and optical communication.

**Author Contributions:** Writing—original draft preparation, T.W.; writing—review and editing, M.H.; conceptualization, T.W., T.O. and M.H.; methodology, T.W. and T.O.; formal analysis and investigation, T.W.; resources, T.W.; data curation, T.W.; supervision and project administration, M.H., T.O. and K.Y. All authors have read and agreed to the published version of the manuscript.

**Funding:** This research was funded by JSPS KAKENHI grunt number JP18H01902, JST through the establishment of university fellowship for the creation of science and technology innovation grant number JPMJFS2130. And the APC was funded by JPMJFS2130.

**Institutional Review Board Statement:** Not applicable.

**Informed Consent Statement:** Not applicable.

**Data Availability Statement:** Data will be made available upon request.

**Conflicts of Interest:** The authors declare no conflict of interest.

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
