# Peer review of "Determining the Optical Geometry of a Gold Semi-Shell under the Kretschmann Configuration"

_photonics, doi:10.3390/photonics10111228_

Round 1

Reviewer 1 Report

Comments and Suggestions for Authors

In the manuscript “Determining the Optical Geometry of a Gold Semi-Shell under the Kretschmann Configuration”, the authors studied the semi-metal shell orientation theoretically and experimentally. The topic is interesting, and the related results can be potentially important to related research. But I have a few comments and questions.

1The language should be examined thoroughly. For example, on page 1 “Recently, plasmonic nanostructures have been used in diverse applications, including as metamaterials [3–8], as well as for nonlinear optical effects [9,10], optothermal and photoelectric conversion [11,12], and biosensing [13–16].”

2: On page 1 “In particular, half-shell and semi-shell structures involving dielectric nanoparticles coated with metals are attracting attention as composite materials with optical anisotropy and large absorption cross-sectional areas [17–19].” What is the difference between half-shell and semi-shell?

3: On page 1 “Using nanoparticles as sources, the resonance wavelength can be adjusted by changing the particle size. Reducing the size to approximately 100 nm can enable the excitation of plasmon modes in the optical wavelength range (visible and near-infrared regions). Therefore, these materials are advantageous for realizing functions within the optical wavelength range.” These sentences are redundant. Why people should care more about the nanoparticles having resonance in the optical wavelength range?

4: On page 2 “Although determining the orientation of the structure is crucial when using dielectric nanoparticles with sizes of approximately 100 nm, observations through scanning electron microscopy (SEM) are challenging owing to charging.” Could more explanation be added?

5: On page 2 “Notably, surface plasmons allow higher-sensitivity signal detection than conventional microscopy”. This sentence should explained in more detail.

6: On page 3 “The wavelength spectrum of evanescent light exhibits a strong bias owing to the excitation of propagating surface plasmons.” What does the strong bias present? More rationale should be added for the 70-degree incident angle selection.

7: In the simulation part, the metal region is not symmetric. Should there not be a variable of angle included? The light incident direction should be specified in the simulation. And are there any experimental results to make the values of the simulation parameters (h1, h2, g) scientific solid?

8: I think Figure 5 is misleading. I suggest putting the simulation results spectra and field distribution together.

9: How many particles were measured based on this method? Since there is no second method to measure the orientation of semi-metal shells. The actual orientation should not be simply divided into these 4 configurations as in the simulation work.

Comments on the Quality of English Language

The language can be improved.

Reviewer 2 Report

Comments and Suggestions for Authors

The manuscript addresses experimentally the determination of the optical geometry of a gold semi-shell under the Kretschmann configuration. The authors have developed a surface-plasmon-based measurement method under Kretschmann configuration, and determined the geometrical orientation of a gold semi-shell structure on a gold thin film using optical measurements only. It is shown that the changes in the scattering light spectrum of a single gold semi-shell can be experimentally observed at wavelengths ranging from 530–700 nm, which reasonably agrees with numerically simulated results. The investigations may have potential applications in optical imaging and sensing.

 I have several remarks as follows.

1) What is the size limitation in determining the optical geometry of a gold semi-shell structure by using this measurement method? The authors need to provide some numerical simulations at least in order to discuss this issue.

2) It seems possible to develop a similar method to determine the optical geometry of a single metallic semi-shell structure on the metal/dielectric nanostructured layers (to replace the Kretschmann configuration), where the surface plasmons can be generated without Kretschmann configuration, for example to see Nature 424, 824 (2003) by Barnes et al., Phys. Rev. Lett. 94, 033903 (2005) by Fan et al, and Phys. Rev. B 76, 195405 (2007) by Tang et al.The authors should add to discuss this issue based on those published works.

This manuscript is well-written. I think this manuscript may present an interesting experimental approach, and it can be published in Photonics after major revision.

Reviewer 3 Report

Comments and Suggestions for Authors

The work titled 'Determining the Optical Geometry of a Gold Semi-Shell under the Kretschmann Configuration' by Watanabe et al., presents an interesting plasmonics technology for demonstrating the Kretschmann configuration to experimentally and numerically show the optical determination of single-structure orientations through a finite-difference time-domain method. This interesting work may be considered for publication provided the authors address the below mentioned comments:

1. Typically SPR occurs for a perticular polarized component of light. How about the polarization selectivity of the proposed platform?

2. The motivation for choosing the semi-shell platform is not clearly elaborated.

3. It is important to discuss the reasons for angular and wavelength dependence from plasmonics perspective. How about spectral resolution and background suppression?

4. The experimental characterization for semi-shell is missing. Please show TEM or EDAX data to validate the chemical composition.

5. What is the outcome that authors anticipate with the use of other metals for making the semi-shells and the platform. Please discuss.

6. The relevant works in this domain should be used to enrich the discussion section: Micromachines 2023, 14(3), 668; J. Chem. Phys. 153, 101101 (2020).

7. The future scope of the work should be discussed from surface plasmon resonance (SPR) and surface plasmon-coupled emission (SPCE) perspective using recent relevant works in this domain, to the broad audience of photonics.

Comments on the Quality of English Language

Needs some improvement.

Round 2

Reviewer 1 Report

Comments and Suggestions for Authors

In the manuscript “Determining the Optical Geometry of a Gold Semi-Shell under the Kretschmann Configuration”, the authors studied the semi-metal shell orientation theoretically and experimentally. And in the revised manuscript, the authors have answered the questions well. The manuscript is qualified for publication.

Reviewer 2 Report

Comments and Suggestions for Authors

The revised manuscript has been improved. As I mentioned before, this work presents an interesting experimental approach. I think it can be considered to be published in Photonics.

Reviewer 3 Report

Comments and Suggestions for Authors

Authors address the reviewer comments well.

Comments on the Quality of English Language

Needs some improvement.